# Resilience and Positive Wellbeing Experienced by 5–12-Year-Old Children with Refugee Backgrounds in Australia: The Childhood Resilience Study

**DOI:** 10.3390/ijerph21050627

**Published:** 2024-05-15

**Authors:** Elisha Riggs, Deirdre Gartland, Josef Szwarc, Mardi Stow, Georgia Paxton, Stephanie J. Brown

**Affiliations:** 1Intergenerational Health, Murdoch Children’s Research Institute, Melbourne, VIC 3052, Australia; elisha.riggs@mcri.edu.au (E.R.); stephanie.brown@mcri.edu.au (S.J.B.); 2Department of General Practice, University of Melbourne, Melbourne, VIC 3052, Australia; 3Department of Pediatrics, University of Melbourne, Melbourne, VIC 3052, Australia; 4Victorian Foundation for Survivors of Torture, Melbourne, VIC 3056, Australia; szwarcj@foundationhouse.org.au (J.S.); stowm@foundationhouse.org.au (M.S.); 5Immigrant Health Service, Royal Children’s Hospital, Melbourne, VIC 3052, Australia; georgia.paxton@rch.org.au

**Keywords:** refugee, resilience, child, positive wellbeing, mental health

## Abstract

Refugee research tends to be deficit based and focused on the risks threatening positive adaptation and wellbeing. High rates of mental (and physical) health issues have been reported for refugee adults and children, including intergenerational trauma. This study uses the new Child Resilience Questionnaire (CRQ), co-designed with refugee background communities, to describe resilience and positive wellbeing experienced by children of refugee-background. The Childhood Resilience Study (CRS) recruited 1132 families with children aged 5–12 years in Victoria and South Australia, Australia. This included the recruitment of 109 families from 4 refugee background communities: Assyrian Chaldean (Iraq, Syria), Hazara (Afghanistan), Karen (Burma, Thailand) and Sierra Leonean families. CRQ-parent/caregiver report (CRQ-P/C) scores were categorised into ‘low’, ‘moderate’ and ‘high’. The child’s emotional and behavioural wellbeing was assessed with the Strengths and Difficulties Questionnaire, with positive wellbeing defined as <17 on the total difficulties score. Tobit regression models adjusted for a child’s age. The CRQ-P/C scores were not different for boys and girls of refugee background. Children of refugee-background (n = 109) had higher average CRQ-P/C scores than other CRS children (n = 1023) in the personal, school and community domains, but were lower in the family domain. Most children with ‘high’ resilience scores had positive wellbeing for both children of refugee-background (94.6%) and other CRS children (96.5%). Contrary to common stereotypes, children of refugee-background show specific individual, family, school and cultural strengths that can help them navigate cumulative and complex risks to sustain or develop their positive wellbeing. A better understanding as to how to build strengths at personal, family, peer, school and community levels where children are vulnerable is an important next step. Working in close collaboration with refugee communities, schools, policy makers and key service providers will ensure the optimal translation of these findings into sustainable practice and impactful public policy.

## 1. Introduction

According to the United Nations High Commission for Refugees (UNHCR), the world is witnessing the highest level of forced population displacement on record. At the end of 2022, there were an estimated 108.4 million people forcibly displaced worldwide, including 35.3 million formally recognised as refugees by the UN. Forced displacement occurs because of persecution, conflict, violence, human rights violations and natural or man-made disasters. Importantly, around half of all people forcibly displaced are children under the age of 18 years [1]. The Refugee Convention set out the international legal definition of a refugee as a person who (a) has a well-founded fear of being persecuted for reasons of race, religion, nationality, membership of a particular social group or political opinion; (b) is outside the country of their nationality; and (c) is unable or, owing to such fear, is unwilling to avail themselves of the protection of that country [2]. This study uses the term ‘refugee background’ to encompass people covered by the above definition and others who have experienced persecution, significant discrimination and human rights abuses in their country of nationality or usual residence and immediate family members (such as children). This expanded and holistic view recognises the unique and often traumatic refugee experiences that people have survived, irrespective of their visa or migration status [3]. Children with parents of refugee background born in Australia are also considered to be of ‘refugee background’, given the potential for the intergenerational transmission of trauma and ongoing challenges of settlement in a new country [4,5]. 

The refugee experience is characterised by a great deal of change, loss and social adversity. When families are forced to flee from their home country due to war, conflict, human rights abuses, and other atrocities, it is common for them to spend months or years in countries of asylum, refugee camps or detention centres, where access to shelter, food, water, employment and education are limited or non-existent, and daily life is a struggle. Many children and their families witness or experience violence and are separated from family and friends. For children and adolescents, all of this occurs during key developmental periods, with potentially significant implications for physical, social, emotional and cognitive outcomes [6,7,8,9,10]. High rates of mental (and physical) health issues have been reported for refugees (both adults and children), particularly post-traumatic stress disorder [10,11,12,13]. Additionally, intergenerational trauma can negatively impact children, including those born in settlement countries [3,5,7]. There is also growing evidence that post-settlement factors are strongly related to mental health disorders for refugee adults [14]. Settlement factors can include protracted asylum-seeking processes/temporary visas, mandatory detention, limited work rights, underemployment, poverty, family separation and grief, racism, social exclusion, status/role changes and adapting to new languages and social customs [14,15,16]. Many settlement factors are intersectional and shaped by the host countries’ political and/or social structures, which can entrench or reduce social inequalities for marginalised communities [17]. 

In response, refugee health research has tended to be dominated by a deficit-based discourse and has often focused on physical or mental health during transition and early settlement. “While this research has been invaluable in understanding the varied experiences and outcomes for refugee-background families, it has neglected a large piece of the settlement and adaptation experience—that of resilience” (p. 678, [18]). Resilience is now predominantly viewed as a dynamic process by which individuals draw on personal characteristics and resources in their environment to successfully navigate adversity over their life course [19]. Individual, relational and contextual factors that support resilience in children have been identified, for example, optimism, social skills, a close bond with a caregiver, school engagement and cultural and community connections [20]. Resilience factors can interact and change over time and by context. For example “Irrespective of culture and context, stable and nurturing relationships are found to support development of various individual resilience assets and provide access to a variety of contextual resources” (p. 1375, [21]). However, it is important not to “romanticise how colonised and marginalised populations adapt to and recover from difficult conditions” (p. 24, [22]). As described above, many of the settlement challenges for refugee families arise from the social and political settings in host countries. Whilst we are exploring the resilience and vulnerability experienced by children of refugee background, we note that these concepts are situated within “cultural, economic and sociopolitical dimensions, all of which are influenced by the dominant political powers and are often linked to each other in times of crisis” (p. 31, [17]).

Resilience has been observed in children of refugee background who show positive personal, social, academic and/or developmental outcomes despite the adversities they have faced in their refugee and settlement journeys [7,9,18,23]. However, our synthesis of the existing child resilience literature identified very limited examinations of resilience in children of refugee background; in addition, the available literature showed a strong focus on personal skills such as the capacity to self-regulate or utilise adaptive coping strategies [18,24]. Beyond the personal realm, relational factors have been shown to be associated with positive outcomes in refugee populations (predominantly for adolescents), including the family parenting style, positive family relationships, and teacher and peer support [8,9,18]. Jafari et al. [18] highlights three significant limitations in the existing literature on resilience in children and adolescents of refugee background: firstly, the focus on resilience as an individual characteristic rather than a “complex transactional process” encompassing “individual strengths, supportive relationships, and cultural values, as well as practices and community resources” (p. 689); secondly, the lack of culturally appropriate tools to measure child and youth resilience; and finally, the very limited conceptualisation of ‘culture’ in the literature as religious and/or spiritual factors alone. 

These limitations mean that in practice, child resilience has most commonly been identified by proxy, often using the Strengths and Difficulties Questionnaire, which is a measure of emotional and behavioural wellbeing developed in a majority population and translated for use with other language groups [25]. Such research is in effect using positive emotional and behavioural wellbeing to identify resilient individuals. More recent research positions resilience as the process of accessing strengths and resources to mediate the relationship between risk/adversity and positive mental health. To provide examples at both ends of an adversity spectrum, exposure to significant, long-lasting or repeated adversity is likely to impact an individual’s mental health, even with access to strengths and resources. Conversely, where adversity exposure has been slight or non-existent, individuals may have good mental health despite having little access to strengths or resources. A better understanding and measurement of the strengths and resources that support good mental health for children of refugee background is vital. Interventions to strengthen their access to such strengths and resources can have both immediate and long-term benefits over their life course [18,23]. 

This paper draws on the data collected in the Childhood Resilience Study (CRS), a project which aimed to improve the available evidence and understanding of resilience in children, beginning with the development of a culturally and socially inclusive multidomain measurement tool. The new Child Resilience Questionnaire (CRQ) was codesigned with Aboriginal and/or Torres Strait Islander communities and refugee-background communities. The tool assesses the strengths and resources within the child and in their family, school, peers and community/culture that can be accessed when adversity arises. This paper draws on the CRQ data collected from over 1000 parents/caregivers of a child aged 5–12 to (1) describe the personal, family, school, peer and community strengths experienced by children of refugee-background (n = 109) compared to other children in the CRS study (n = 1023); (2) identify gender differences in resilience scores of children of refugee background; and (3) examine the relationship between refugee background children’s resilience scores and their emotional/behavioural wellbeing.

## 2. Materials and Methods

### 2.1. Study Design and Context

The study aims of the Childhood Resilience Study originated from our previous work with Aboriginal/Torres Strait Islander and refugee-background communities, who wanted to better understand how some children did well, while others in similar situations did not [24,26]. We employed a strengths-based approach, with community consultation and bilateral knowledge exchange underpinning the design and conduct of the study. The study details have been published elsewhere. Briefly, the study was grounded in community-based participatory methods and comprised a systematic review and co-design of the content, including collaborative item and scale development [24]. Families from diverse cultural and social backgrounds were recruited from a range of sources for two rounds of psychometric testing, including outpatient clinics in a large tertiary hospital, two existing longitudinal cohort studies, plus community-based recruitment of Australian Aboriginal and Torres Strait Islander and families of refugee background [26,27]. 

The CRQ assesses the resources within the self, family, school, peers and community/culture that a child can access in times of adversity. A parent/caregiver report version (CRQ-PC) [26] and child report version (CRQ-C) [27] have been published. A school report version is currently in development (CRQ-S). 

Our partnership with the Victorian Foundation for the Survivors of Torture (Foundation House) was established at the inception of the Child Resilience Study and underpinned all engagement with the refugee-background families [24,26,27]. Throughout the study, the research partnership maintained a commitment to consultation, engagement and co-design and oversaw all fieldwork with the families of refugee background. Overall, the Child Resilience Questionnaire (CRQ) was co-designed and psychometrically tested using input from almost 200 parents and 200 children who were recruited from five refugee-background communities: Assyrian Chaldean families from Iraq and Syria, Hazara families from Afghanistan, Karen families from Burma and Thailand, Tamil families from Sri Lanka and Sierra Leonean families from Sierra Leone. (The processes for working with Aboriginal families are described elsewhere [28]).

In this paper, the parent report data (CRQ-PC) are used to describe the strengths and vulnerabilities experienced by refugee-background children, including a comparison with the strengths/resources reported for other children in the Childhood Resilience Study.

### 2.2. Recruitment

Diverse families with children aged 5–12 years were recruited to the Childhood Resilience Study from September 2017–March 2020 via the four sources described below, with one adult per family completing a CRQ-parent/caregiver report (CRQ-P/C) about their child:Urban and rural-based families with diverse economic, cultural and social backgrounds were recruited via outpatient clinics in a large tertiary children’s hospital (n = 460). To compare the modes of administration for psychometric testing, parents/caregivers were randomised to complete the CRQ-P/C on an iPad or paper using a random number generator.Families of refugee background were recruited via networks of community researchers and completed the CRQ-P/C on an iPad or paper as preferred (n = 109).Aboriginal families were recruited via community networks of Aboriginal investigators and researchers and completed the CRQ-P/C on an iPad or paper as preferred (n = 68). The CRQ-P/C was also included in the wave 2 paper questionnaire for an Aboriginal cohort study (Aboriginal Families Study) of children aged 5–8 years (n = 231).Families were recruited via a population-based pregnancy cohort study of 1507 mothers and their first child, which was followed up over 10 years (Maternal Health Study). Mothers with multiple children were invited to complete an online REDCap CRQ-P/C survey about a child other than the cohort child. The children were aged 5–9 years (n = 264).

All of these families were recruited as part of the second round of psychometric testing of the CRQ; the recruitment processes and details are available elsewhere [27,28]. 

The four refugee-background communities described in this paper reflected the backgrounds and community networks of our community researchers and represent both more recently arrived and more established refugee communities including Assyrian Chaldean (from Iraq and Syria), Hazara (from Afghanistan), Karen (from Burma and Thailand) and Sierra Leonean communities. The community researchers participated in ‘in-house’ training on research processes, including the steps involved in gaining informed consent, protocols for supporting participants if they became distressed and self-care during fieldwork. Recruitment and data collection were conducted at locations determined by the community researchers and included schools, community events and community venues. Parents/caregivers and children completed the questionnaires on an iPad or paper as preferred, in English, Karen, Arabic or Dari, with assistance from the community researchers as needed. The iPad version included an audio recording of the questionnaire in four languages to support participants who did not have good English literacy. 

### 2.3. Measures

***Child resilience*** was assessed using the Child Resilience Questionnaire-P/C report. There are 11 scales across the socioecological domains of personal, family, school and community (see scales and item examples in text Box 1). Higher scores indicate the child has access to a greater number of resilience resources when challenges arise. The stem question is “How often are the following true for your child”, with response options of 0 “Not at all”, 1 “Not often”, 2 “Sometimes”, 3 “Most of the time” and 4 “All of the time”. 

The mean scale, domain and total scores were calculated. Established cutoff scores are not yet available, so the total CRQ-P/C score were divided into tertiles (thirds) to represent low, medium and high resilience scores respective to the children in the Childhood Resilience Study (n = 1132). 

Box 1CRQ Domains, scales and sample items.

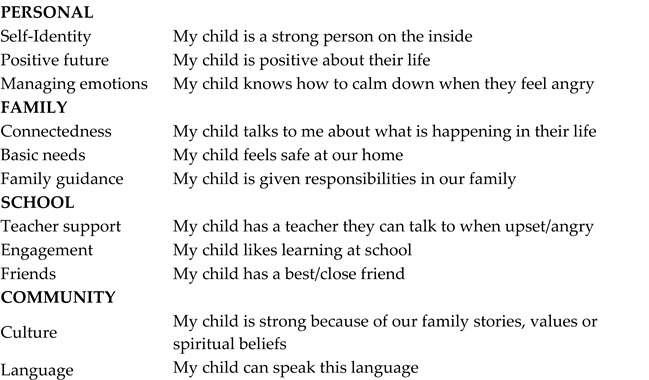



***Emotional and behavioural wellbeing***: The Strengths and Difficulties Questionnaire (SDQ) is a measure of emotional and behavioural wellbeing for children aged 4–16 years and has been used with refugee-background populations, with noted limitations [29]. Participants were asked to rate 25 attributes as 1 “Not true”, 2 “Somewhat true” or 3 “Certainly true”. Five subscales are scored (prosocial behaviour, emotional symptoms, conduct problems, hyperactivity/inattention and peer relationships problems). A total difficulties score is derived from the four subscales, excluding prosocial behaviour, to indicate the level of risk for emotional and/or behavioural problems. 

The original 5 factor solution has not been supported in studies with participants of refugee background [29,30,31]. Furthermore, SDQ items have been identified as potentially not working in different cultural contexts [29,30,31,32]. For example, Dang and Nguyen [31] report that the item “Gets along better with adults than children” (translated into Vietnamese) was influenced by cultural values and did not discriminate between Vietnamese children who had been psychiatrically referred and children in the control group. It has also been recommended to rely on ‘local’ cutoffs when working with refugee-background communities [29]. A cutoff of 17 on the SDQ total difficulties score is recommended to identify emotional and/or behavioural difficulties for Australian children [33]. We adopted a strengths-based approach that has been used with Australian Aboriginal children (who also experienced significant social adversity and intergenerational trauma), where a child scoring <15 was considered to have positive emotional/behavioural wellbeing, henceforth ‘positive wellbeing” [34].

***Sociodemographic characteristics***: In addition, brief data were collected on a range of sociodemographic factors. The Working Group and community researchers advised researchers to ask for minimal identifying data to avoid any concerns participants might have in sharing personal information given their refugee background. Accordingly, we only asked about the relationship of parent–caregiver participant to the child, gender, age, country of birth and year of arrival in Australia. 

### 2.4. Analyses

The family characteristics are presented for families of refugee background and other Childhood Resilience Study families. Pearson Chi^2^ tests were used to identify statistical differences in low, medium and high CRQ-P/C total scores by family characteristics. Regression models were used to examine the differences in the CRQ-P/C scores, comparing (1) children of refugee background and other children in the Childhood Resilience Study and (2) child gender. Tobit linear regression models were used with CRQ-P/C scale, domain and total scores as a more robust approach for censored data, given the ceiling effects observed in some scores [35]. As there is a significant developmental span from 5 to 12 years and a strong potential for the school resilience factors (and potentially others) to be affected by how long a child has attended school, all regression models were adjusted for the child’s age. 

## 3. Results

### 3.1. Participants

A total of 109 refugee background parent/caregivers of children aged 5–12 years completed the CRQ-P/C. The characteristics of the parent/caregivers and their child are described in Table 1. Most participants were the parent of the child they were reporting on (94.4%). A higher proportion of mothers/female caregivers participated (65.7%), but one in three were fathers/male caregivers. The parents/caregivers reported on an even number of male and female children, and the children spanned all ages (mean age = 9.1, SD = 2.2). Half of the families had four or more children (i.e., 52.2% of the target children had three or more siblings). The average age of the children reported on was 9.1 years. Almost a third of the children were born in Australia, just over a third had been living in Australia for 1–3 years and the remaining children had lived in Australia for more than 4 years. The four communities involved in the study were evenly represented.

For the other families in the Childhood Resilience Study, most participants were parents of the child they were reporting on (97.4%). Almost all the children were born in Australia (96.1%). A high proportion of mothers/female caregivers participated (87.1%) (Note: this was skewed through recruitment via two mother–child cohorts). Most families had 1–2 children (74.0%), and the average age of the children was 8.9 years. Most children had been born in Australia (96.1%).

### 3.2. Resilience in Children of Refugee Background 

Over half of the children of refugee background were categorised as having high resilience scores, scoring in the top third of CRQ-P/C scores (52.3%). As shown in Table 1, there were no differences in the sociodemographic characteristics of children who were scored as having low, medium and high resilience scores, with the exception of the country of birth. Over half of the children of refugee background in the ‘high’ resilience score category had been born in Australia (63.6%) compared to children who had been in Australia for 0–3 years (55.3%) or 4–6 years (31.6%) (Chi^2^ = 8.0, *p* = 0.018). 

The mean CRQ-P/C scale and domain scores by child gender are presented in Figure 1. There were no gender differences in the mean CRQ-P/C scale or domain scores for girls of refugee background compared to boys (see also Appendix A). 

As shown in Figure 2 (with the full data presented in Table 2), differences were observed in the mean CRQ-P/C scale or domain scores, with children of refugee background scoring higher on average on 5 of the 11 subscales (and 3 of 4 domains) compared to other Childhood Resilience Study children but scoring lower on the family domain. 

As reported by their parent/caregiver, children of refugee background had significantly higher *personal strengths* domain scores than other Childhood Resilience Study children (13.0 and 11.7, respectively). That is, children of refugee background had CRQ-P/C *personal strengths* domain scores 1.2 points higher on average after adjusting for child age (Adj β= 1.2, 95%CI 0.7–1.8). This difference was evident in two of the three personal strengths subscales: the *positive self-identity* (Adj β = 1.6, 95%CI 1.1, 2.2) and *emotion regulation* subscales (Adj β = 0.9, 95%CI 0.4–1.5). There was no statistical difference in the positive *future* subscale, although children of refugee background were also scored higher.

In the *school* domain, on average, children of refugee background were scored 1.7 points higher on the *school engagement* subscale compared to the other Childhood Resilience Study children (Adj β = 1.7, 95%CI 1.1–2.4). While there were no differences in the *teacher support* or *friend* subscales, children of refugee background were scored higher on the overall *school strengths* domain score, although the difference was small 562(Adj β = 0.9, 95%CI 0.4–1.5).

In the *community* domain, children of refugee background scored higher on average on both subscales—*connectedness to culture* (Adj β = 1.4, 95%CI 0.6–2.1) and *connectedness to language* (Adj β = 2.5, 95%CI 2.0–3.0)—compared to other Childhood Resilience Study children. Overall, refugee background children scored, on average, 3.0 points higher in the *community strengths* domain than other Childhood Resilience Study children.

In contrast, the refugee background children’s *family strengths* domain score was, on average, slightly lower than the other Childhood Resilience Study children after adjusting for age (Adj β = −0.5, 95%CI −1.0,−0.0). Specifically, children of refugee background were scored lower on average on the *basic needs* (Adj β = −1.3, 95%CI −1.9,−0.8) and *family guidance* subscales (Adj β = −0.6, 95%CI −1.2–−0.1) than the other Childhood Resilience Study children. There was no difference in the *family connectedness* subscale score. 

There was no difference in the overall CRQ-P/C score for children of refugee background compared to the other Childhood Resilience Study children.

### 3.3. Resilience and Emotional and Behavioural Wellbeing

The majority of children of refugee background (85.1%) were classified as having positive emotional/behavioural wellbeing (SDQ total difficulties score < 15). Most children (94.6%) in the high resilience score category had positive emotional/behavioural wellbeing. However, as only 16 children scored as not having positive wellbeing, this resulted in very small numbers of children without across the different levels of resilience scores. Thus, it can be seen in Figure 3 that wellbeing is related to the resilience scores, but there is no consistent trend for high to low resilience scores. 

A more consistent trend of a decreasing proportion of children with positive wellbeing for high to low categories of resilience scores was observed in the larger sample of other Childhood Resilience Study children. Almost all children with high resilience scores showed positive wellbeing (96.5%). 

A slightly lower proportion of children (92.5%) in the medium resilience score category had positive wellbeing, dropping to 67.8% for children with low resilience scores.

## 4. Discussion

Despite histories inclusive of adversity and trauma, children of refugee background displayed a range of strengths in socioecological domains, supporting their positive wellbeing. Half of the children of refugee background aged 5–12 years were categorised as having ‘high resilience scores’ on the CRQ-P/C, as reported by their parents/caregivers. Compared to other children in the Childhood Resilience Study, children of refugee background had higher mean resilience domain scores for *personal*, *school* and *community* strengths, and lower mean scores for *family* strengths. Specifically, children of refugee background scored higher than other children in the Childhood Resilience Study on a range of subscales: *positive self-identity, emotion regulation, school engagement, connectedness to culture* and *language*. In contrast, the resilience scores were lower for the children of refugee background on the *family guidance* and *basic needs* subscales (there was no difference in terms of *family connectedness*). No differences were observed for refugee-background girls compared to boys on any resilience domain or subscale scores, nor in terms of positive wellbeing. 

Despite experiences of significant adversity and challenges, just over half the children of refugee background showed positive wellbeing. An examination of the resilience scores showed that the wellbeing of children was buffered where they had high access to strengths and resources (high resilience scores), with 95% showing positive wellbeing (SDQ < 15). Similar findings have been reported in another Australian study of 43 newly arrived refugee children. The majority (63%) of children with four or more protective factors (e.g., father present on arrival, pre-settlement parent education, having relatives in Australia prior to arrival, proximity to own ethnic community and parent employment) had SDQ total difficulties scores in the ‘normal’ range. Furthermore, having more protective factors increased the children’s likelihood of stable or improved SDQ scores from two to three years after arrival. Modifiable post-arrival factors included stability in the children’s school and residence, parental employment, financial and marital stability, proximity to one’s own ethnic community and external community support [36].

In the current study, between a third and a fifth of the children with medium or low resilience scores did not have positive wellbeing, as identified by SDQ total difficulties score of <15. These children do not appear to have the range or specific strengths and resources needed to protect their wellbeing in the context of past or current adversity/challenges experienced. Additionally, the children with poor wellbeing (including in the ‘high resilience score category’) may have experienced or continue to experience more extensive adversity and trauma that is impacting their wellbeing, despite their access to strengths.

Strong ties to family and community have consistently been reported as a key protective factor for children’s wellbeing [20,23,36]. The uniqueness of the CRQ is the capacity to explore specific strengths and vulnerabilities. While the children of refugee background scored high on community/cultural subscales compared to the other children, they were scored lower on the *basic needs* and *family guidance* scales. The *basic needs* scale has items such as “My child feels safe at our home” and “My child has their own space in the place where we live”. Items in the family guidance scale include “Our family has routines” and “My child helps with things like shopping”. However, perhaps most importantly, these aspects of family life did not appear to be impacting the children’s *connectedness to family*, as reported by caregivers (e.g., my child talks to me about their feelings). Discrimination and socioeconomic disadvantage is a common, if not universal, settlement experience across different sociopolitical, cultural and economic settings [12,14,17]. Related factors such as poor living conditions, social isolation, lack of education and/or employment rights, underemployment, racism and acculturation issues can significantly impact caregivers’ mental health and capacity to provide for their children [14,15,16]. Parent/caregiver mental and physical health may also be a factor in refugee caregivers’ capacity to parent [9], with a range of prior migration and settlement experiences underpinning poorer health outcomes for refugee families [3,13]. Post-traumatic stress disorder is more prevalent in refugee compared to non-refugee children and adults, with implications for parenting [8,11,13,37]. However, similar to non-refugee populations, anxiety and depression are the most common mental health diagnoses [13]. Parental support and linkages to health and social services during the antenatal and early childhood period are likely to set up a stable platform of support for parents and children [38], while access to early education and school settings will similarly benefit their children [12,39]. However, a greater understanding and responsiveness to the role of intergenerational trauma within families and communities of refugee background, mapped to intervention programs that support the whole family, are vital to reduce the hardship and daily struggles that can continue to play a role despite settlement (or birth) in a high-income country [5,12].

In this study, a slightly higher proportion of children of refugee background who were born in Australia had ‘high’ resilience scores (63.6%) compared to the children born overseas (47.4%), as reported by their parent/caregiver. Although second generation children can experience intergenerational trauma and other family settlement hardships, they do not directly experienced the upheaval, traumatic experiences and displacement of forced migration (although children of refugee background born in Australia may have experienced time in detention centres), all factors that are likely to have significant and lasting impacts at home, school and community settings for overseas-born children [3,5,40,41,42].

It is also important to acknowledge the great diversity in forced migration and settlement experiences associated with different refugee backgrounds. The research findings reported in this paper will reflect the communities we engaged with, and further research is required to understand the resilience of children specific to different refugee experiences and backgrounds.

### Strengths and Limitations

The Child Resilience Questionnaire offers a new approach to measure resilience in children, including refugee-background children, highlighting personal strengths and resources within the family, school, peers and community/culture. Much of the research examining resilience in children of refugee background to date has focused on personal strengths, used qualitative methods and/or proxy measures of resilience and has typically been conducted with children living in refugee camps or other temporary settings in low- and middle-income countries. Whilst these studies have vital and important findings, our study is the first to utilise a co-designed, multi-domain quantitative measure of childhood resilience. It is also the first study to focus on children of refugee background living in a high-income country, including children born in Australia and overseas. This is a positive step in the inclusion and recognition of the diversity within communities of refugee background. 

A strength of this study was the diversity of communities included in the co-design of the CRQ and in this paper. We included both communities that are well established in Melbourne, Australia, and more recently arrived communities. However, we acknowledge the great diversity that exists for refugee-background communities, families and individuals in their forced displacement, migration and settlement experiences, culture and language, so these findings may not be generalisable for other communities or families. Furthermore, we have small numbers in each community and for the study overall, although 109 is higher than many existing refugee studies. 

## 5. Conclusions

Children of refugee background may be exposed to cumulative and complex risks. This study provides evidence for significant personal, family, school and community/cultural strengths that are buffering their positive wellbeing. Children of refugee background who have access to a range of individual, family and social support factors can navigate risks to ‘regain, sustain or grow’ their mental health [43]. This is contrary to the common discourse and important for families, schools, communities and wider society to recognise and celebrate.

The suggestion of weaker scores for family strengths, and more specifically, family guidance and basic needs, reflects some of the pressures on families and communities transitioning to life in a new country and the legacy of forced displacement. Where children of refugee background do not have access to specific resilience strengths and resources, there is an important policy imperative to redirecting health and social care efforts in this direction. 

High-level evidence with economic analyses is required to develop and evaluate universal and targeted interventions to benefit refugee families and communities. School settings, as a universal service, provide an opportunity to identify child/class/school level strengths and vulnerabilities that can guide interventions and resource investments around specific personal, peer, school and community strengths that buffer children’s outcomes. Fundamental to this work will be working in close collaboration with refugee communities, schools, policy makers and key service providers to ensure the optimal translation of these findings into sustainable practice and impactful public policy.

## Figures and Tables

**Figure 1 ijerph-21-00627-f001:**
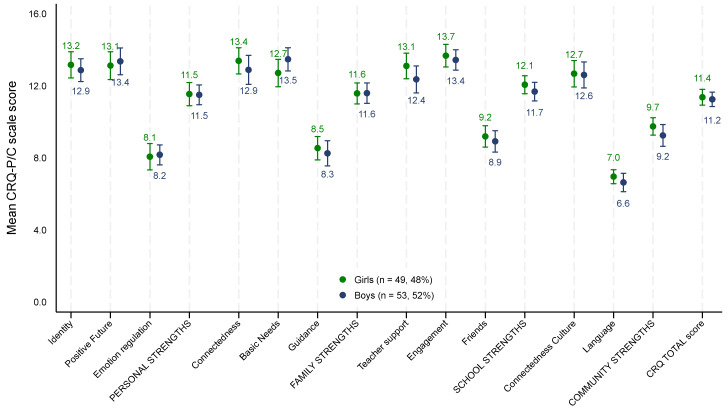
Mean CRQ-P/C scale and domain scores, with 95% confidence intervals for children of refugee background by caregiver-reported gender (n = 109).

**Figure 2 ijerph-21-00627-f002:**
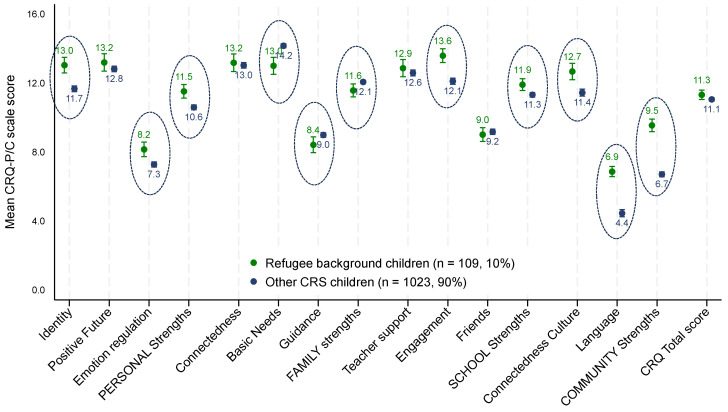
Mean CRQ-P/C scale and domain scores, with 95% confidence intervals for the children of refugee background and other Childhood Resilience Study children. Circles indicate statistically significant differences after adjusting for the child’s age.

**Figure 3 ijerph-21-00627-f003:**
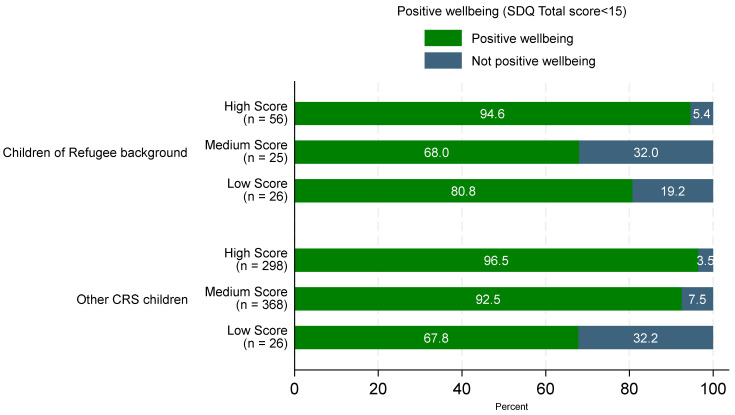
Proportion of children with positive emotional and/or behavioural wellbeing by high, medium and low resilience score categories for children of refugee background (n = 109) and other Childhood Resilience Study children (n = 1023).

**Table 1 ijerph-21-00627-t001:** Family characteristics of other CRS families and families of refugee background and resilience scores (n = 109).

	Other Childhood Resilience Study Families ^1^	Refugee Background-Families	CRQ-P/C Resilience Categories for Children of Refugee Background			Pearson
			Low Score	Medium Score	High Score	Chi^2^
	n(%)	n(%)	n(%)	n(%)	n(%)	*p*-Value
Participant report						
Gender						
Female	886 (87.1)	71 (65.7)	16 (22.5)	15 (21.1)	40 (56.3)	0.416
Male	131 (12.9)	37 (34.3)	10 (27)	11 (29.7)	16 (43.2)	
Relationship to child						
Mother/father	993 (97.4)	102 (94.4)	26 (25.5)	23 (22.5)	53 (52)	0.191
Family caregiver/guardian	27 (2.6)	6 (5.6)	0 (0)	3 (50)	3 (50)	
Gender of child						
Female	471 (46.9)	49 (48.0)	12 (24.5)	11 (22.4)	26 (53.1)	0.971
Male	534 (53.1)	53 (52.0)	14 (26.4)	12 (22.6)	27 (50.9)	
Child age mean (SD)	8.9 (2.3)	9.1 (2.2)	9.9 (2.0)	8.7 (2.4)	9.0 (2.3)	
5–6 years	210 (20.6)	19 (17.4)	2 (10.5)	8 (42.1)	9 (47.4)	0.269
7–8 years	255 (25)	19 (17.4)	3 (15.8)	3 (15.8)	13 (68.4)	
9–10 years	193 (19.0)	36 (33.0)	10 (27.8)	8 (22.2)	18 (50.0)	
11–12 years	360 (35.4)	35 (32.1)	11 (31.4)	7 (20)	17 (48.6)	
Number of siblings						
None	47 (5.0)	2 (2.2)	0 (0)	0 (0)	2 (100)	0.201
1–2	693 (74.0)	41 (45.6)	13 (31.7)	9 (22)	19 (46.3)	
3–4	154 (16.5)	37 (41.1)	8 (21.6)	12 (32.4)	17 (45.9)	
>4	42 (4.5)	10 (11.1)	0 (0)	2 (20)	8 (80)	
Child country of birth						
Australia	977 (96.1)	33 (30.3)	1 (3)	11 (33.3)	21 (63.6)	
Overseas	40 (3.9)	76 (69.7)	25 (32.9)	15 (19.7)	36 (47.4)	
Administration method						
iPad	228 (22.3)	24 (22.0)	3 (12.5)	5 (20.8)	16 (66.7)	0.224
Paper	544 (53.2)	85 (78.0)	23 (27.1)	21 (24.7)	41 (48.2)	
Online (REDCap)	251 (24.5)					
Years lived in Australia						
Born in Australia		33 (31.7)	1 (3.0)	11 (33.3)	21 (63.6)	0.030
0–3 years		38 (36.5)	10 (26.3)	7 (18.4)	21 (55.3)	
4–6 years		19 (18.3)	8 (42.1)	5 (26.3)	6 (31.6)	
7+ years		14 (13.5)	5 (35.7)	3 (21.4)	6 (42.9)	
Refugee Community						
Assyrian Chaldean (Iraq, Syria)		29 (26.6)	10 (34.5)	6 (20.7)	13 (44.8)	0.097
Hazara (Afghanistan)		30 (27.5)	3 (10)	8 (26.7)	19 (63.3)	
Karen (Burma, Thailand)		28 (25.7)	10 (35.7)	8 (28.6)	10 (35.7)	
Sierra Leone (Sierra Leone)		22 (20.2)	3 (13.6)	4 (18.2)	15 (68.2)	
	1023 (100)	109 (100)	26 (23.9)	26 (23.9)	57 (52.3)	

^1^ Excludes children of refugee background (n = 109).

**Table 2 ijerph-21-00627-t002:** Mean CRQ-P/C domain and scale scores for children of refugee background and the other Childhood Resilience Study children, with Tobit logistic regression modelling differences (n = 229).

Domain		Other Childhood Resilience Study Children (n = 1023)	Children of Refugee Background (n = 109)	Tobit Regression	
CRQ Scale	Items (Range)	Mean [95%CI]	Mean [95%CI]	Adj.β ^1^ [95%CI]	*p*-Value
PERSONAL strengths					
Positive self-identity	4 (0–16)	11.7 [11.5–11.8]	13.0 [12.6–13.5]	1.6 [1.1–2.2]	<0.001
Positive future	4 (0–16)	12.8 [12.6–13.0]	13.2 [12.7–13.7]	0.6 [−0.0–1.3]	0.065
Emotion regulation	3 (0–12)	7.3 [7.1–7.4]	8.2 [7.7–8.6]	0.9 [0.4–1.5]	<0.001
*Mean personal domain score*	*11 (0–16)*	*10.6 [10.5–10.7]*	*11.5 [11.1–11.9]*	*1.2 [0.7–1.8]*	*<0.001*
FAMILY strengths					
Connectedness	4 (0–16)	13.0 [12.9–13.2]	13.2 [12.7–13.7]	0.3 [−0.3–1.0]	0.299
Basic needs	4 (0–16)	14.2 [14.0–14.3]	13.0 [12.5–13.5]	−1.3 [−1.9–−0.8]	<0.001
Guidance	3 (0–12)	9.0 [8.8–9.1]	8.4 [8.0–8.9]	−0.6 [−1.2–−0.1]	0.020
*Mean family domain score*	*11 (0–16)*	*12.1 [12.0–12.2]*	*11.6 [11.2–11.9]*	*−0.5 [−1.0–−0.0]*	*0.011*
SCHOOL strengths					
Teacher support	4 (0–16)	12.6 [12.4–12.8]	12.9 [12.4–13.4]	0.3 [−0.6–1.2]	0.499
School engagement	4 (0–16)	12.1 [11.9–12.3]	13.6 [13.2–14.0]	1.7 [1.1–2.4]	<0.001
Friends	3 (0–12)	9.2 [9.0–9.3]	9.0 [8.6–9.4]	−0.2 [−0.9–0.4]	0.425
*Mean school domain score*	*11 (0–16)*	*11.3 [11.2–11.5]*	*11.9 [11.6–12.2]*	*0.9 [0.4–1.5]*	*0.003*
COMMUNITY strengths					
Cultural connectedness	4 (0–16)	11.4 [11.2–11.6]	12.7 [12.2–13.1]	1.4 [0.6–2.1]	<0.001
Connectedness to language ^2^	4 (0–8)	4.4 [4.2–4.7]	6.9 [6.6–7.2]	2.5 [2.0–3.0]	<0.001
*Mean community domain score*	*8 (0–16)*	*6.7 [6.6–6.9]*	*9.5 [9.2–9.9]*	*3.0 [2.5–3.5]*	*<0.001*
Total RESILIENCE score					
*Mean total scale score*	43 (0–16)	11.1 [11.0–11.2]	11.3 [11.0–11.6]	0.4 [0.0–0.8]	0.061

^1^ Adjusted for child’s age. ^2^ Completed only for children who spoke more than one language.

## Data Availability

The conditions of ethics approval preclude data sharing via a public repository. However, the Investigator Team welcome inquiries about the data and proposals for collaboration. Interested parties are invited to contact the lead investigators Elisha Riggs (elisha.riggs@mcri.edu.au), Deirdre Gartland (deirdre.gartland@mcri.edu.au) or Stephanie Brown (stephanie.brown@mcri.edu.au).

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
