# Peer review of "Resilience and Positive Wellbeing Experienced by 5–12-Year-Old Children with Refugee Backgrounds in Australia: The Childhood Resilience Study"

_ijerph, 2024, doi:10.3390/ijerph21050627_

Round 1
Reviewer 1 Report
Comments and Suggestions for Authors
Thank you very much for focusing on such an important topic and the attempt to deviate from a deficit approach in reflecting on wellbeing of children with migrate background. While the resilient approach is radically different from a deficit model, it has important shortcomings which links it with neoliberalist perspectives that value individual success and focus on individuals rather than the structural shortcomings. This puts a heave pressure on marginalized communities, and could potentially lead to blaming already traumatized groups for yet another failure for not creating a nurturing environment at home.
While this research research, starts nicely with critically reflecting on approaches to define resilience, it is crucial to also reflect on "resilience" as a construct and the approaches that glorify that as opposed to critically reflecting on the structural flaws of societies and their inadequacies to create policies and opportunities where the "margin comes to the center".
Here are some literature that might help in reflecting on the notion of resilience critically:
Sou, G. (2022). Reframing resilience as resistance: Situating disaster recovery within colonialism. The Geographical Journal, 188(1), 14-27.
Shearer Demir, H. (2023). Vulnerability and Resilience. In: Displacement Governance and the Illusion of Integration. Springer, Cham. https://doi.org/10.1007/978-3-031-38655-
Reviewer 2 Report
Comments and Suggestions for Authors
The article entitled “Resilience and Positive Wellbeing Experienced by Children of Refugee Background Aged 5-12 Years in Australia: The Childhood Resilience Study” aims at investigating resilience levels and positive wellbeing of 109 refugee parent/caregivers of children aged 5-12 years coming from four different refugees’ communities.
I have appreciated the manuscript that, in my opinion, gives an important contribution to the international literature on the resilience of children with refugee background. I think the topic is urgent and the paper covers a very timely argument.
I believe the paper is well written and the methodology is appropriate with the objectives of the study. I agree with the authors when they stated that one (in my opinion the principal) limitation of the study regards that the diversity of communities included. The investigation the authors carried out, in fact, put together a wide range of participants with very different background. I believe that you should stress the need for further studies able to discriminate between these backgrounds as well as to discriminate between pre-migratory experiences which might be different and depend on the country of provenience. I think that the research within the field of forced migration can really benefit from a culturally sensitive perspective, and this should be achieved even more.
See for example:
Tessitore, F., Parola, A., Margherita, G. (2023). Mental Health Risk and Protective Factors of Nigerian Male Asylum Seekers Hosted in Southern Italy: a Culturally Sensitive Quantitative Investigation. Journal of Racial and Ethnic Health Disparities, 2023, 10(2), pp. 730–742
Taking also into consideration the paper I above suggested, I believe that some integrations might be added to the Introduction section. Here, the authors argued that the refugee health research has tended to be dominated by a deficit-based discourse and this is indubitably true. However, a multidimensional approach to the refugees’ experiences is increasing in literature even more and many studies are focusing on both protective as well as risk factors for asylum seekers and refugees’ mental health (maybe most of these are on refugees’ adults). I would evaluate to deepen this aspect in order to offer a comprehensive discussion on the state of art of literature on this theme.
Furthermore, I would ask to the authors to explain well the reasons that supported the recruitment (the inclusion criteria). For example, why children aged between 5-12? I understood that the study is part of a wider project, however, I believe that the project had some inclusion criteria purposefully chosen.
I would also suggest revising the abstract. In my opinion, this is quite chaotic, and the main issues of the study and its objectives have the risk to not be understood.
In conclusion, I would also suggest authors to add within the future directions the need to increase qualitative studies on this topic to explore in-depth the different levels of resilience and their interconnections with the family’s background.
Round 2
Reviewer 1 Report
Comments and Suggestions for Authors
Thank you for the changes to the manuscript. The additional notes to the paper present sufficient explanations in response to my critiques.